# Synthesis and Anticancer Evaluation of 4-Anilinoquinolinylchalcone Derivatives

**DOI:** 10.3390/ijms24076034

**Published:** 2023-03-23

**Authors:** Cheng-Yao Yang, Min-Yu Lee, Yeh-Long Chen, Jun-Ping Shiau, Yung-Hsiang Tsai, Chia-Ning Yang, Hsueh-Wei Chang, Chih-Hua Tseng

**Affiliations:** 1Department of Medicinal and Applied Chemistry, College of Life Science, Kaohsiung Medical University, Kaohsiung City 80708, Taiwan; 2Graduate Institute of Medicine, College of Medicine, Kaohsiung Medical University, Kaohsiung 80708, Taiwan; 3Division of Breast Oncology and Surgery, Department of Surgery, Kaohsiung Medical University Hospital, Kaohsiung Medical University, Kaohsiung 80708, Taiwan; 4Department of Seafood Science, National Kaohsiung University of Science and Technology, Kaohsiung 811213, Taiwan; 5Institute of Precision Medicine, National Sun Yat-sen University, Kaohsiung 80424, Taiwan; 6Department of Biomedical Science and Environmental Biology, College of Life Science, Kaohsiung Medical University, Kaohsiung 80708, Taiwan; changhw@kmu.edu.tw; 7Center for Cancer Research, Kaohsiung Medical University, Kaohsiung 80708, Taiwan; 8Department of Fragrance and Cosmetic Science, College of Pharmacy, Kaohsiung Medical University, Kaohsiung City 80708, Taiwan; 9School of Pharmacy, College of Pharmacy, Kaohsiung Medical University, Kaohsiung City 80708, Taiwan; 10Department of Medical Research, Kaohsiung Medical University Hospital, Kaohsiung City 80708, Taiwan; 11Department of Pharmacy, Kaohsiung Municipal Ta-Tung Hospital, Kaohsiung City 80145, Taiwan; 12College of Professional Studies, National Pingtung University of Science and Technology, Pingtung County 912301, Taiwan

**Keywords:** 4-anilinoquinolinylchalcone, lapatinib, cytotoxicity, reactive oxygen species, apoptosis

## Abstract

A series of 4-anilinoquinolinylchalcone derivatives were synthesized and evaluated for antiproliferative activities against the growth of human cancer cell lines (Huh-7 and MDA-MB-231) and normal lung cells (MRC-5). The results exhibited low cytotoxicity against human lung cells (MRC-5). Among them, (*E*)-3-{4-{[4-(benzyloxy)phenyl]amino}quinolin-2-yl}-1-(4-methoxyphenyl) prop-2-en-1-one (**4a**) was found to have the highest cytotoxicity in breast cancer cells and low cytotoxicity in normal cells. Compound **4a** causes ATP depletion and apoptosis of breast cancer MDA-MB-231 cells and triggers reactive oxygen species (ROS)-dependent caspase 3/7 activation. In conclusion, it is worth studying 4-anilinoquinolinylchalcone derivatives further as new potential anticancer agents for the treatment of human cancers.

## 1. Introduction

Cancer is a disease that occurs when changes in a group of normal cells within the body lead to uncontrolled, abnormal growth forming a lump called a tumor [1]; this is true of all cancers except leukemia (cancer of the blood). If left untreated, tumors can grow and spread into the surrounding normal tissue or to other parts of the body via the bloodstream and lymphatic systems and can affect the digestive, nervous, and circulatory systems or release hormones that may affect body function. Liver and breast cancers are common malignancies worldwide and the leading causes of cancer-induced death [2].

Chalcones are natural products that display various biological activities, including antimalarial, anti-protozoal, anti-inflammatory, anti-depressant, and anticancer [3,4,5,6,7,8,9,10,11,12]. Because of their abundance in plants and ease of synthesis, the chalcone class of compounds has attracted extensive studies. Chalcones, which demonstrated potential in vitro and in vivo activity against drug-susceptible and drug-resistant cancers, are useful templates for developing novel anticancer agents [8]. On the other hand, the quinoline skeleton is one of the critical building elements for many natural and synthetic heterocycles, which possess a wide variety of biological effects such as bactericidal, antitumor, anti-inflammatory, and antiviral activities [12,13,14,15,16,17,18].

Hybrid molecules not only have more favorable properties such as enhanced activity and improved specificity but also could overcome drug resistance, so hybridization of chalcone moiety with other anticancer pharmacophores represents a promising strategy to develop novel anticancer agents with high efficacy [19]. In recent years, numerous chalcone-quinoline hybrids have been prepared and evaluated for their anticancer activities. Some exhibited excellent in vitro and in vivo potency, revealing their potential as putative anticancer drugs [10]. In our previous study, we have synthesized a series of chalcone derivatives in which an aryl moiety was replaced with a quinoline nucleus and evaluated for their biological activities [20,21,22]. Among them, (*E*)-3-[3-(4-methoxyphenyl)quinolin-2-yl]-1-phenylprop-2-en-1-one (1) was active against the growth of H1299 and SKBR-3 with IC_50_ values of 1.41 and 0.70 μM, respectively [21]. 

Lapatinib, a tyrosine kinase dual inhibitor targeting epidermal growth factor receptors (EGFR) and Her2/Neu, has been widely accepted in ongoing preclinical or clinical applications for the treatment of various solid tumors, including those of the breast, lung, liver, head and neck, colon, prostate, gastric and ovarian cancers [23,24,25,26]. Abo-Zeid et al. demonstrated that lapatinib exhibited cytotoxicity on the MDA-MB-231 cell line with an IC_50_ value of 32.5 μM after 24 h of treatment [27]. Chen et al. have also revealed IC_50_ values of 2.11, 3.42, and 4.85 μM, respectively, for the lapatinib-treated Huh-7, HepG2, and HA22T cancer cell lines [27]. To further explore 4-anilinoquinolinylchalcone derivatives as potential anticancer agents, a series of novel target compounds (Figure 1) were designed and synthesized. We considered the hybrid approach utilizing the chalcone structure of compound-1 and Phenylether amine substructure from lapatinib. The target structures are more drug-like with a shape similar to the compact cis-form of Combretastatin A4. They also possessed suitable molecular weights and the number of hydrogen-bond donating/accepting groups according to Lipinski’s rule of five. These target compounds were evaluated for their antiproliferative activities against the growth of Huh-7 and MDA-MB-231 cancer cell lines and MRC-5 (normal human lung cells), and the results are outlined in Table 1.

## 2. Results and Discussions

### 2.1. Chemistry

(*E*)-3-(4-Chloroquinolin-2-yl)-1-(4-methoxyphenyl)prop-2-en-1-ones (**2**) and (*E*)-3-(4-chloroquinolin-2-yl)-1-(4-fluoroxyphenyl)prop-2-en-1-ones (**3**), respectively, were reacted with substituted anilines to give the target 4-anilinoquinolinylchalcone derivatives (**4a**–**h** and **5a**–**h**), respectively, (Figure 1) as described in our previous report [18]. The structure of **4a**–**5h** was determined by NMR (^1^H and ^13^C) (spectra data can be found in Appendix A) and further confirmed by elemental analysis. 

### 2.2. Inhibition of Cell Proliferation

All the synthesized 4-anilinoquinolinylchalcone derivatives were evaluated in vitro against two cancer cells (Huh-7 and MDA-MB-231) and a non-cancer normal fibroblast lung cell line (MRC-5) using an XTT (sodium 3′-[1-(phenylamino-carbonyl)-3,4-tetrazolium}-bis(4-methoxy-6-nitro)benzene sulfonic acid hydrate) assay. The concentration that inhibited the growth of 50% of cells (IC_50_) was determined from the linear portion of the curve by calculating the concentration of the tested agent that reduced absorbance in treated cells, compared to control cells, by 50%. The IC_50_ results of these 4-anilinoquinolinylchalcone derivatives are summarized (Table 1). All 4-anilinoquinolinylchalcone derivatives were cytotoxic to all cancer cell lines (IC_50_ < 2.03 μM) and non-cytotoxic against normal cells (MRC-5) (IC_50_ > 20 μM). The 4-methoxyphenyl derivatives (R_1_ = OMe, **4a**–**h**) are more cytotoxic than 4-fluorophenyl derivatives (R_1_ = F, **5a**–**h**), indicating that an electron-donating substituent is more active than that of an electron-withdrawing group. For the 4-methoxyphenyl derivatives (R_1_ = OMe, **4a**–**h**), an electron-withdrawing group substituted at the benzyloxy-phenyl moiety (**4d**, R_2_ = 3-Cl, IC_50_ = 0.69 μM) is more active than unsubstituted (**4a**, R_2_ = H, IC_50_ = 1.47 μM) or the electron-donating group (**4f**, R_2_ = 3-OMe, IC_50_ = 1.41 μM) against the Huh-7 cell line. The same trend was observed in which compound **4d** (IC_50_ = 0.18 μM) was more active than **4f** (IC_50_ = 1.94 μM) against the MDA-MB-231 cell line. (*E*)-3-{4-{[4-(Benzyloxy)phenyl]amino} quinolin-2-yl}-1-(4-methoxyphenyl)prop-2-en-1-one (**4a**) and its substituted derivatives, **4d** and **4e**, exhibited comparable cytotoxicity against both cancer cells, and **4a** was selectively active against the growth MDA-MB-231, with an IC_50_ value of 0.11 μM. Thus, **4a** was selected as a lead compound for further mechanism studies.

Some quinoline derivatives also show antiproliferative function against cancer cells. For example, 2,9-bis [2-(pyrrolidin-1-yl)ethoxy]-6-{4-[2-(pyrrolidin-1-yl)ethoxy]phenyl}-11*H*-indeno[1,2-*c*]quinolin-11-one (BPIQ) inhibits human retinoblastoma Y79 cell proliferation, i.e., IC_50_ = 13.64 μM at 48 h XTT assay [28]. 9-[3(dimethylamino)propoxy]-6-{4-[3-dimethylamino)propoxy]phenyl}-2-fluoro-11*H*-indeno[1,2-*c*]quinolin-11-one (DFIQ) shows antiproliferation against non-small-cell lung A549 cancer cells, i.e., IC_50_ = 3.53 μM at 48 h trypan blue exclusion assay [29]. Similarly, chalcone derivatives such as *S17* exhibit antiproliferative effects on gastric cancer MGC803 cells, i.e., IC_50_ = 6.75 μM at 48 h MTT assay [30]. Since the 4-anilinoquinolinylchalcone derivative as listed (Table 1) contains both quinoline and chalcone structures, these quinoline and chalcone derivative studies [28,29] may partly explain the antiproliferative effects of these derivatives.

### 2.3. Downregulation of ATP by Compound ***4a*** against Breast Cancer Cells

ATP levels are another cell viability indicator [31,32,33]. To further confirm the inhibition of the cell proliferative ability of compound **4a**, the cellular ATP levels of breast cancer MDA-MB-231 cells were assessed (Figure 2). Like the XTT assay, **4a** showed dose-responsive downregulation of ATP in breast cancer cells.

### 2.4. Upregulation of Apoptosis by Compound ***4a*** against Breast Cancer Cells

To examine the apoptosis-inducing ability of compound **4a**, the annexin V/7-aminoactinomycin D (7-AAD) change in breast cancer MDA-MB-231 cells was assessed. Compound **4a** showed more significant apoptosis of breast cancer cells at 5 and 10 μM than control and 1 μM (Figure 3). Similarly, some quinoline and chalcone derivatives, such as BPIQ [28] and *S17* [30], show apoptosis induction. Compound **4a,** consisting of both versatile quinoline and chalcone moieties, was also active in inducing cancer cell apoptosis.

### 2.5. Reactive Oxygen Species (ROS) Were Induced by Compound ***4a*** in Breast Cancer Cells

To evaluate the ROS-inducing ability of compound **4a**, the flow cytometry-based ROS detection of breast cancer MDA-MB-231 cells was conducted. Compound **4a** showed dose-responsive induction of ROS of breast cancer cells (Figure 4).

Similarly, some quinoline and chalcone derivatives show the function of cellular ROS induction. For example, BPIQ, a quinoline derivative, induces ROS production in retinoblastoma cells [28]. (*E*)-3-(3,5-dimethoxyphenyl)-1-(1-hydroxynaphthalen-2-yl) prop-2-en-1-one can evoke ROS burst in rheumatoid arthritis fibroblast-like synoviocytes [34]. The chalcone derivative *S17* shows ROS generation in gastric cancer cells [30]. Since compound **4a** contains both quinoline and chalcone structures, these characteristics partly explain its ROS induction ability**.**

### 2.6. Upregulation of Caspase 3/7 Activity by Compound ***4a*** against Breast Cancer Cells

Since apoptosis and ROS induction was observed, the role of ROS in apoptosis signaling is worth investigating its ability to trigger apoptosis. This potential role of ROS was examined by the pretreatment of *N*-acetylcysteine (NAC), a ROS inhibitor [35,36]. Several anticancer studies, such as psoralidin [37], nitrated [6,6,6]tricycles compound SK1 [38], and *Aaptos suberitoid* extract [39], have demonstrated that NAC can reverse the ROS-induced apoptosis in the example of caspase 3/7 activation.

Using the caspase 3/7 luminescence assay, the modulating ability of breast cancer MDA-MB-231 cells on caspase 3/7 signaling with compound **4a** was evaluated. Like annexin V-detected apoptosis (Figure 3), compound **4a** was inactive at the concentration of 1 μM. However, it exhibited a significant caspase 3/7 activation of breast cancer cells at 5 and 10 μM (Figure 5). Moreover, this caspase 3/7 activation with compound **4a** was suppressed by NAC pretreatment. These data suggest that compound **4a** activates caspase 3/7 of breast cancer cells in an ROS-dependent manner.

## 3. Materials and Methods

### 3.1. Chemistry

Melting points were determined on an Electrothermal IA9100 (Electrothermal, Staffordshire, UK) melting point apparatus and were uncorrected. Nuclear magnetic resonance (^1^H and ^13^C) spectra were recorded on a Varian-Unity-400 spectrometer (Varian, Palo Alto, CA, USA). Chemical shifts were expressed in parts per million (δ) with tetramethylsilane (TMS) as an internal standard. Thin-layer chromatography was performed on silica gel 60 F-254 plates purchased from E. Merck and Co (Darmstadt, Germany). The elemental analyses were performed in the Instrument Center of the National Science Council at National Cheng-Kung University or National Taiwan University using Heraeus CHN-O Rapid EA (Heraeus, Waltham, MA, USA), and all values are within ± 0.4% of the theoretical compositions.


**(*E*)-3-{4-{[4-(Benzyloxy)phenyl]amino}quinolin-2-yl}-1-(4-methoxyphenyl)prop-2-en-1-one**
**
*(*
**
**4a**
**
*)*
**


The suspension of compound **2** (2.0 mmol) and 4-(benzyloxy)aniline (2.0 mmol) were dissolved in 30 mL EtOH, and the added 6N HCl (1 mL) was refluxed for 8 h (TLC monitoring). After cooling, it evaporated under reduced pressure to give solid compounds. The products were purified by recrystallization from EtOH to give compound **4a**. Yield 87% orange solid. Mp 200–201 °C. ^1^H NMR (400 MHz, DMSO-*d_6_*) δ 3.88 (s, 3H, OMe), 5.19 (s, 2H, CH_2_), 7.11–7.23 (m, 5H, Ar-H and 3-H), 7.37–7.51 (m, 7H, Ar-H), 7.66 (d, 1H, *J* = 16.0 Hz, CH = CH), 7.75–7.79 (m, 1H, 6-H), 8.02–8.06 (m, 1H, 7-H), 8.29 (d, 2H, *J* = 8.8 Hz, Ar-H), 8.58 (d, 1H, *J* = 8.0 Hz, 5-H), 8.75 (d, 1H, *J* = 8.4 Hz, 8-H), 8.88 (d, 1H, *J* = 16.0 Hz, CH = CH), 10.99 (s, 1H, NH). ^13^C and DEPT NMR (100 MHz, DMSO-*d_6_*) δ 55.69 (OCH_3_), 69.62 (OCH_2_), 102.15 (CH), 114.23 (2CH), 115.94 (2CH), 116.60 (CH), 120.49 (CH), 123.38 (CH), 126.89 (CH), 126.94 (2CH), 127.83 (2CH), 128.00 (CH), 128.52 (2CH), 129.52 (C), 129.74 (C), 131.63 (2CH), 131.80 (CH), 133.93 (CH), 134.05 (CH), 136.80 (C), 139.10 (C), 147.29 (C), 155.15 (C), 157.47 (C), 163.89 (C), 186.45 (C). Anal. calcd for C_32_H_26_N_2_O_3_·1.4HCl: C 71.47, H 5.15, N 5.21; found: C 71.54, H 5.25, N 5.16.


**(*E*)-3-{4-{{4-[(2-Chlorobenzyl)oxy]phenyl}amino}quinolin-2-yl}-1-(4-methoxy**
**-**
**phenyl) prop-2-en-1-one *(*4b*)***


From **2** and 4-[(2-chlorobenzyl)oxy]aniline, as described for **4a**: Yield 89% an orange solid. Mp 160–161°C. ^1^H NMR (400 MHz, DMSO-*d_6_*) δ 3.89 (s, 3H, OMe), 5.24 (s, 2H, CH_2_), 7.12 (d, 2H, *J* = 8.8 Hz, Ar-H), 7.21–7.25 (m, 3H, Ar-H and 3-H), 7.42–7.56 (m, 5H, Ar-H and CH = CH), 7.65–7.78 (m, 3H, Ar-H and 6-H), 8.01–8.05 (m, 1H, 7-H), 8.27 (d, 2H, *J* = 8.8 Hz, Ar-H), 8.53 (d, 1H, *J* = 8.8 Hz, 5-H), 8.74 (d, 1H, *J* = 8.4 Hz, 8-H), 8.82 (d, 1H, *J* = 16.0 Hz, CH = CH), 10.90 (br s, 1H, NH), 14.62 (br s, 0.6 H, HCl). ^13^C NMR (100 MHz, DMSO-*d_6_*) δ 55.69, 67.25, 102.02, 114.27 (2C), 115.92 (2C), 116.74, 120.84, 123.25, 126.89 (2C), 127.43, 129.46 (2C), 129.56, 130.03, 130.15, 130.32, 131.36, 131.57 (2C), 132.72, 133.97, 134.06, 134.34, 139.47, 147.67, 154.96, 157.19, 163.89, 186.60. Anal. calcd for C_32_H_25_ClN_2_O_3_·0.5HCl: C 71.26, H 4.78, N 5.20; found: C 71.19, H 4.96, N 5.13.


**(*E*)-3-{4-{{4-[(2-Fluorobenzyl)oxy)]phenyl}amino}quinolin-2-yl}-1-(4-methoxy-phenyl)**
**prop-2-en-1-one (4c)**


From **2** and 4-[(2-fluorobenzyl)oxy]aniline, as described for **4a**: Yield 92% as an orange solid. Mp 175–176 °C. ^1^H NMR (400 MHz, DMSO-*d_6_*) δ 3.89 (s, 3H, OMe), 5.22 (s, 2H, CH_2_), 7.12 (d, 2H, *J* = 8.8 Hz, Ar-H), 7.20–7.31 (m, 5H, Ar-H and 3-H), 7.44–7.49 (m, 3H, Ar-H), 7.60–7.64 (m, 1H, 6-H), 7.67 (d, 1H, *J* = 16.0 Hz, CH = CH), 7.75–7.69 (m, 1H, Ar-H), 8.02–8.06 (m, 1H, 7-H), 8.28 (d, 2H, *J* = 8.8 Hz, Ar-H), 8.56 (d, 1H, *J* = 8.8 Hz, 5-H), 8.75 (d, 1H, *J* = 8.4 Hz, 8-H), 8.86 (d, 1H, *J* = 16.0 Hz, CH = CH), 10.96 (br s, 1H, NH), 14.54 (br s, 1 H, HCl). ^13^C NMR (100 MHz, DMSO-*d_6_*) δ 55.70, 63.93 (*J* = 3.1 Hz), 102.06, 114.20, 114.28 (2C), 115.46 (*J* = 20.4 Hz), 115.89 (2C), 116.65, 123.30, 123.56, 124.60 (*J* = 3.8 Hz), 126.98 (2C), 129.54, 129.97, 130.60 (*J* = 7.5 Hz), 130.88, 130.92, 131.61 (2C), 131.73 (*J* = 24.3 Hz), 134.07 (*J* = 7.6 Hz), 139.15, 141.54, 147.47, 155.20, 157.30, 160.49 (*J* = 244.8 Hz), 163.92, 186.56. Anal. calcd for C_32_H_25_FN_2_O_3_·1.2HCl: C 70.08, H 4.83, N 5.11; found: C 69.69, H 4.87, N 4.92.


**(*E*)-3-**
**{4-{{4-[(3-Chlorobenzyl)oxy]phenyl}amino}quinolin-2-yl}-1-(4-methoxy-phenyl) prop-2-en-1-one (*4d*)**


From **2** and 4-[(3-chlorobenzyl)oxy]aniline, as described for **4a**: Yield 81% an orange solid. Mp 132–133 °C. ^1^H NMR (400 MHz, DMSO-*d_6_*) δ 3.87 (s, 3H, OMe), 5.18 (s, 2H, CH_2_), 7.10–7.16 (m, 5H, Ar-H and 3-H), 7.38–7.47 (m, 5H, Ar-H), 7.56–7.62 (m, 3H, 6-H and CH = CH), 7.76–7.80 (m, 1H, 7-H), 8.01 (d, 1H, *J* = 8.4 Hz, 5-H), 8.08–8.16 (m, 3H, Ar-H and CH = CH), 8.45 (d, 1H, *J* = 8.4 Hz, 8-H), 9.30 (s, 1H, NH). ^13^C NMR (100 MHz, DMSO-*d_6_*) δ 55.62, 68.55, 101.44, 114.22 (2C), 115.76 (2C), 118.74, 122.17, 125.49 (2C), 126.25, 126.58, 127.37, 127.80, 128.08, 130.11, 130.42 (2C), 130.62, 131.02 (2C), 132.60, 133.14, 139.69, 142.23, 147.21, 150.46, 152.23, 155.48, 163.44, 187.68. Anal. calcd for C_32_H_25_ClN_2_O_3_·0.5HCl: C 71.26, H 4.78, N 5.19; found: C 71.19, H 4.84, N 5.16.


**(*E*)-3-{4-{{4-[(3-Fluorobenzyl)oxy]phenyl}amino}quinolin-2-yl}-1-(4-methoxy-phenyl) prop-2-en-1-one (4e)**


From **2** and 4-[(3-fluorobenzyl)oxy]aniline, as described for **4a**: Yield 95% an orange solid. Mp 171–172 °C. ^1^H NMR (400 MHz, DMSO-*d_6_*) δ 3.89 (s, 3H, OMe), 5.22 (s, 2H, CH_2_), 7.10–7.25 (m, 6H, Ar-H and 3-H), 7.32–7.36 (m, 2H, Ar-H), 7.46–7.49 (m, 3H, Ar-H), 7.67 (d, 1H, *J* = 15.6 Hz, CH = CH), 7.75–7.79 (m, 1H, 6-H), 8.02–8.06 (m, 1H, 7-H), 8.28 (d, 2H, *J* = 8.0 Hz, Ar-H), 8.55 (d, 1H, *J* = 8.4 Hz, 5-H), 8.75 (d, 1H, *J* = 8.0 Hz, 8-H), 8.86 (d, 1H, *J* = 15.6 Hz, CH = CH), 10.98 (br s, 1H, NH), 14.65 (br s, 1H, HCl). ^13^C NMR (100 MHz, DMSO-*d_6_*) δ 55.71, 68.71 (*J* = 1.5 Hz), 102.04, 114.30 (2C), 114.31 (*J* = 21.2 Hz), 114.71 (*J* = 21.3 Hz), 116.01 (2C), 116.63, 120.48, 123.31, 123.62 (*J* = 2.3 Hz), 126.99 (2C), 129.54, 129.90, 130.56 (*J* = 8.3 Hz), 131.62 (2C), 131.93, 133.94, 134.19, 139.05, 139.78 (*J* = 7.6 Hz), 147.42, 155.25, 157.25, 162.23 (*J* = 242.5 Hz), 163.95, 186.56. Anal. calcd for C_32_H_25_FN_2_O_3_·1.5HCl: C 68.70, H 4.78, N 5.00; found: C 68.31, H 4.99, N 4.93.


**(*E*)-3-{4-{{4-[(3-Methoxybenzyl)oxy]phenyl}amino}quinolin-2-yl}-1-(4-methoxy**
**-**
**phenyl)prop-2-en-1-one**
**
*(*
**
**4f*)***


From **2** and 4-[(3-methoxybenzyl)oxy]aniline, as described for **4a**: Yield 85% a yellow solid. Mp 110–111 °C. ^1^H NMR (400 MHz, DMSO-*d_6_*) δ 3.78 (s, 3H, OMe), 3.87 (s, 3H, OMe), 5.12 (s, 2H, CH_2_), 6.90–6.93 (m, 1H, Ar-H), 7.05–7.14 (m, 7H, Ar-H and 3-H), 7.31–7.38 (m, 3H, Ar-H), 7.53–7.55 (m, 1H, 6-H), 7.59 (d, 1H, *J* = 15.6 Hz, CH = CH), 7.73–7.77 (m, 1H, 7-H), 7.96 (d, 1H, *J* = 8.4 Hz, 5-H), 8.04–8.08 (m, 3H, Ar-H and CH = CH), 8.42 (d, 1H, *J* = 8.4 Hz, 8-H), 9.08 (s, 1H, NH). ^13^C NMR (100 MHz, DMSO-*d_6_*) δ 55.04, 55.59, 69.35, 101.35, 113.19, 113.24, 114.19 (2C), 115.71 (2C), 118.99, 119.77, 121.99, 125.19, 125.30 (2C), 125.92, 129.09, 129.56, 130.09, 130.18, 130.90 (2C), 132.76, 138.68, 143.31, 148.26, 149.89, 152.84, 155.51, 159.35, 163.35, 187.85. Anal. calcd for C_23_H_17_NO_2_·0.8HCl: C 72.61, H 5.33, N 5.13; found: C 72.43, H 5.69, N 5.05.


**
*(E*
**
**)-3-{4-{{4-[(4-Chlorobenzyl)oxy]phenyl}amino}quinolin-2-yl}-1-(4-methoxy**
**-**
**phenyl)prop-2-en-1-one *(*4g*)***


From **2** and 4-[(4-chlorobenzyl)oxy]aniline, as described for **4a**: Yield 92% an orange solid. Mp 142–143 °C. ^1^H NMR (400 MHz, DMSO-*d_6_*) δ.89 (s, 3H, OMe), 5.19 (s, 2H, CH_2_), 7.12 (d, 2H, *J* = 8.8 Hz, Ar-H), 7.18–7.22 (m, 3H, Ar-H and 3-H), 7.45–7.54 (m, 6H, Ar-H), 7.66 (d, 1H, *J* = 16.0 Hz, CH = CH), 7.75–7.79 (m, 1H, 6-H), 8.02–8.06 (m, 1H, 7-H), 8.29 (d, 2H, *J* = 9.2 Hz, Ar-H), 8.57 (d, 1H, *J* = 8.8 Hz, 5-H), 8.75 (d, 1H, *J* = 8.4 Hz, 8-H), 8.87 (d, 1H, *J* = 16.0 Hz, CH = CH), 10.97 (br s, 1H, NH), 14.72 (br s, 1 H, HCl). ^13^C NMR (100 MHz, DMSO-*d_6_*) δ 55.70, 68.71, 102.07, 114.28 (2C), 115.98 (2C), 116.64, 120.53, 123.30, 126.95 (2C), 128.49 (2C), 129.53, 129.60 (2C), 129.87, 131.62 (2C), 131.88, 132.53, 133.99, 134.13, 135.87 (2C), 139.12, 147.42, 155.20, 157.27, 163.92, 186.55. Anal. calcd for C_32_H_25_ClN_2_O_3_·1.5HCl: C 66.74, H 4.65, N 4.87; found: C 66.67, H 4.78, N 4.76.


**(*E*)-3-{4-{**
**{**
**4-**
**[**
**(4-Fluorobenzyl)oxy**
**]**
**phenyl}amino}quinolin-2-yl}-1-(4-methoxy**
**-**
**phenyl)prop-2-en-1-one**
**
*(*
**
**4h*)***


From **1** and 4-[(4-fluorobenzyl)oxy]aniline, as described for **4a**: Yield 88% a yellow solid. Mp 176–177 °C. ^1^H NMR (400 MHz, DMSO-*d_6_*) δ 3.87 (s, 3H, OMe), 5.13 (s, 2H, CH_2_), 7.10–7.15 (m, 5H, Ar-H and 3-H), 7.23–7.28 (m, 2H, Ar-H), 7.38 (d, 2H, *J* = 8.8 Hz, Ar-H), 7.53–7.62 (m, 4H, Ar-H and CH = CH and 6-H), 7.74–7.78 (m, 1H, 7-H), 7.98 (d, 1H, *J* = 8.0 Hz, 5-H), 8.06–8.11 (m, 3H, Ar-H and CH = CH), 8.44 (d, 1H, *J* = 8.4 Hz, 8-H), 9.17 (s, 1H, NH). ^13^C NMR (100 MHz, DMSO-*d_6_*) δ 55.58, 68.76, 101.36, 114.17 (2C), 115.24 (2C, *J* = 21.2 Hz), 115.69 (2C), 118.88, 122.05, 125.25, 125.35 (2C), 126.17, 128.68, 130.00 (2C, *J* = 9.3 Hz), 130.15, 130.26, 130.92 (2C), 132.68, 133.28 (*J* = 3.1 Hz), 142.88, 147.85, 150.11, 152.58, 155.51, 161.77 (*J* = 241.8 Hz), 163.36, 187.77. Anal. calcd for C_32_H_25_FN_2_O_3_·0.4HCl: C 74.02, H 4.94, N 5.40; found: C 74.38, H 4.99, N 5.41.


**(*E*)-3-{4-{[4-(Benzyloxy)phenyl]amino}quinolin-2-yl}-1-(4-fluorophenyl)prop-2-en-1-one *(*5a*)***


From **3** and 4-(benzyloxy)aniline, as described for **4a**: Yield 81% an orange solid. Mp 162–163 °C. ^1^H NMR (400 MHz, DMSO-*d_6_*) δ 5.18 (s, 2H, CH_2_), 7.18–7.22 (m, 3H, Ar-H and 3-H), 7.35–7.52 (m, 9H, Ar-H), 7.67–7.73 (m, 2H, CH = CH and 6-H), 7.95–7.98 (m, 1H, 7-H), 8.30–8.33 (m, 2H, Ar-H), 8.39 (d, 1H, *J* = 8.4 Hz, 5-H), 8.62–8.68 (m, 2H, CH = CH and 8-H), 10.45 (s, 1H, NH). ^13^C NMR (100 MHz, DMSO-*d_6_*) δ 69.58, 102.02, 115.90 (2C), 116.09 (2C, *J* = 21.7 Hz), 117.27, 123.01, 126.54 (2C), 127.83 (2C), 127.97, 128.51 (2C), 130.07, 130.50, 132.03 (2C, *J* = 9.6 Hz), 133.18, 133.48 (*J* = 2.7 Hz), 136.88 (2C), 148.55, 153.93, 157.02, 165.43 (*J* = 251.65 Hz), 187.46. Anal. calcd for C_31_H_23_FN_2_O_2_·1.2HCl: C 71.82, H 4.72, N 5.41; found: C 71.74, H 4.78, N 5.29.


**(*E*)-3-{4-{{4-[(2-Chlorobenzyl)oxy]phenyl}amino}quinolin-2-yl}-1-(4-fluorophenyl) prop-2-en-1-one *(*5b*)***


From **3** and 4-[(2-chlorobenzyl)oxy]aniline, as described for **4a**: Yield 84% an orange solid. Mp 192–193 °C. ^1^H NMR (400 MHz, DMSO-*d_6_*) δ 5.25 (s, 2H, CH_2_), 7.23–7.26 (m, 3H, Ar-H and 3-H), 7.41–7.57 (m, 7H, Ar-H), 7.65–7.67 (m, 1H, Ar-H), 7.74 (d, 1H, *J* = 16.0 Hz, CH = CH), 7.77–7.81 (m, 1H, 6-H), 8.03–8.07 (m, 1H, 7-H), 8.34–8.38 (m, 2H, Ar-H), 8.50 (d, 1H, *J* = 8.8 Hz, 5-H), 8.74 (d, 1H, *J* = 8.4 Hz, 8-H), 8.81 (d, 1H, *J* = 16.0 Hz, CH = CH), 10.98 (br s, 1H, NH), 14.61 (br s, 1H, HCl). ^13^C NMR (100 MHz, DMSO-*d_6_*) δ 67.28, 102.10, 115.96 (2C), 116.13 (2C, *J* = 22.0 Hz), 116.69, 120.52, 123.32, 126.97 (2C), 127.06, 127.46, 129.47, 130.01, 130.07, 130.35, 131.51, 132.16 (2C, *J* = 9.9 Hz), 132.75, 133.28 (*J* = 3.0 Hz), 134.06, 134.25, 134.88, 139.08, 147.28, 155.23, 157.31, 165.54 (*J* = 252.4 Hz), 187.15. Anal. calcd for C_31_H_22_ClFN_2_O_2_·1.2HCl: C 67.34, H 4.24, N 5.07; found: C 67.04, H 4.26, N 4.89.


**(*E*)-3-{4-{{4-[(2-Fluorobenzyl)oxy]phenyl}amino}quinolin-2-yl}-1-(4-fluorophenyl) prop-2-en-1-one *(*5c*)***


From **2** and 4-[(2-fluorobenzyl)oxy]aniline, as described for **4a**: Yield 78% an orange solid. Mp 207–208 °C. ^1^H NMR (400 MHz, DMSO-*d_6_*) δ 5.22 (s, 2H, CH_2_), 7.23–7.32 (m, 5H, Ar-H and 3-H), 7.43–7.47 (m, 5H, Ar-H), 7.59–7.64 (m, 1H, Ar-H), 7.73 (d, 1H, *J* = 16.0 Hz, CH = CH), 7.76–7.81 (m, 1H, 6-H), 8.03–8.07 (m, 1H, 7-H), 8.35–8.38 (m, 2H, Ar-H), 8.53 (d, 1H, *J* = 8.4 Hz, 5-H), 8.75 (d, 1H, *J* = 8.4 Hz, 8-H), 8.84 (d, 1H, *J* = 16.0 Hz, CH = CH), 10.99 (br s, 1H, NH), 14.68 (br s, 1H, HCl). ^13^C NMR (100 MHz, DMSO-*d_6_*) δ 63.95 (*J* = 3.9 Hz), 102.16, 115.47 (*J* = 21.0 Hz), 115.91 (2C), 116.12 (2C, *J* = 21.7 Hz), 116.66, 120.49, 123.33, 123.48, 123.62, 124.58 (*J* = 3.4 Hz), 127.01 (2C), 127.06, 129.92, 130.62 (*J* = 8.0 Hz), 130.73 (*J* = 24.0 Hz), 131.55, 132.18 (2C, *J* = 9.6 Hz), 133.29 (*J* = 2.7 Hz), 134.48 (*J* = 8.0 Hz), 139.04, 147.22, 155.31, 157.35, 160.50 (*J* = 244.7 Hz), 165.54 (*J* = 251.2 Hz), 187.13. Anal. calcd for C_31_H_22_F_2_N_2_O_2_·1.5HCl: C 68.03, H 4.34, N 5.11; found: C 68.03, H 4.26, N 4.97.


**(*E*)-3-{4-{{4-[(3-Chlorobenzyl)oxy]phenyl}amino}quinolin-2-yl}-1-(4-fluorophenyl) prop-2-en-1-one (5d*)***


From **3** and 4-[(3-chlorobenzyl)oxy]aniline, as described for **4a**: Yield 75% an orange solid. Mp 172–173 °C. ^1^H NMR (400 MHz, DMSO-*d_6_*) δ 5.22 (s, 2H, CH_2_), 7.21–7.24 (m, 3H, Ar-H and 3-H), 7.41–7.47 (m, 7H, Ar-H), 7.56–7.57 (m, 1H, Ar-H), 7.72 (d, 1H, *J* = 15.6 Hz, CH = CH), 7.76–0-7.80 (m, 1H, 6-H), 8.02–8.07 (m, 1H, 7-H), 8.35–8.38 (m, 2H, Ar-H), 8.53 (d, 1H, *J* = 8.4 Hz, 5-H), 8.74 (d, 1H, *J* = 8.8 Hz, 8-H), 8.83 (d, 1H, *J* = 15.6 Hz, CH = CH), 10.97 (br s, 1H, NH), 14.73 (br s, 1H, HCl). ^13^C NMR (100 MHz, DMSO-*d_6_*) δ 68.62, 102.15, 116.02 (2C), 116.13 (2C, *J* = 21.7 Hz), 116.68, 120.56, 123.31, 126.28, 126.97 (2C), 127.05, 127.39, 127.89, 129.93, 130.46, 131.48, 132.28 (2C, *J* = 9.9 Hz), 133.16, 133.29 (*J* = 2.3 Hz), 134.22, 134.92, 139.13, 139.42, 147.25, 155.22, 157.23, 165.54 (*J* = 251.5 Hz), 187.13. Anal. calcd for C_31_H_22_ClFN_2_O_2_·1.3HCl: C 66.90, H 4.23, N 5.04; found: C 66.86, H 4.17, N 4.94.


**(*E*)-3-{4-{{4-[(3-Fluorobenzyl)oxy]phenyl}amino}quinolin-2-yl}-1-(4-fluorophenyl)prop-2-en-1-one *(*5e*)***


From **3** and 4-[(3-fluorobenzyl)oxy]aniline, as described for **4a**: Yield 74% an orange solid. Mp 213–214 °C. ^1^H NMR (400 MHz, DMSO-*d_6_*) δ 5.22 (s, 2H, CH_2_), 7.17–7.24 (m, 4H, Ar-H and 3-H), 7.32–7.36 (m, 2H, Ar-H), 7.41–7.51 (m, 5H, Ar-H), 7.72 (d, 1H, *J* = 16.0 Hz, CH = CH), 7.75–0-7.79 (m, 1H, 6-H), 8.02–8.06 (m, 1H, 7-H), 8.36–8.39 (m, 2H, Ar-H), 8.56 (d, 1H, *J* = 8.4 Hz, 5-H), 8.76 (d, 1H, *J* = 8.4 Hz, 8-H), 8.87 (d, 1H, *J* = 16.0 Hz, CH = CH), 11.02 (br s, 1H, NH), 14.71 (br s, 1H, HCl). ^13^C NMR (100 MHz, DMSO-*d_6_*) δ 68.71, 102.22, 114.30 (*J* = 22.0 Hz), 114.71 (*J* = 21.3 Hz), 116.00 (2C), 116.21, 116.65, 120.46, 123.35, 123.62 (*J* = 3.1 Hz), 126.99 (2C), 127.03, 129.88, 130.55 (*J* = 8.3 Hz), 131.54, 132.19 (*J* = 9.1 Hz), 133.27, 133.29, 134.22, 134.79, 139.05, 139.74, 139.80, 147.16, 155.28, 157.27, 162.22 (*J* = 241.8 Hz), 165.54 (*J* = 261.7 Hz), 187.09. Anal. calcd for C_31_H_22_F_2_N_2_O_2_·1.9HCl: C 66.26, H 4.30, N 4.99; found: C 65.94, H 4.42, N 4.83.


**(*E*)-1-(4-Fluorophenyl)-3-{4-{{4-[(3-methoxybenzyl)oxy]phenyl}amino}quinolin-2-yl}prop-2-en-1-one *(*5f*)***


From **3** and 4-[(3-methoxybenzyl)oxy]aniline, as described for **4a**: Yield 79% an orange solid. Mp 165–166 °C. ^1^H NMR (400 MHz, DMSO-*d_6_*) δ 3.78 (s, 3H, OMe), 5.16 (s, 2H, CH_2_), 6.91–6.94 (m, 1H, Ar-H), 7.06–7.07 (m, 2H, Ar-H), 7.19–7.23 (m, 3H, Ar-H and 3-H), 7.32–7.36 (m, 1H, Ar-H), 7.42–7.48 (m, 4H, Ar-H), 7.72 (d, 1H, *J* = 16.0 Hz, CH = CH), 7.75–0-7.79 (m, 1H, 6-H), 8.02–8.06 (m, 1H, 7-H), 8.36–8.39 (m, 2H, Ar-H), 8.56 (d, 1H, *J* = 8.4 Hz, 5-H), 8.76 (d, 1H, *J* = 8.4 Hz, 8-H), 8.88 (d, 1H, *J* = 16.0 Hz, CH = CH), 11.00 (br s, 1H, NH), 14.68 (br s, 1H, HCl). ^13^C NMR (100 MHz, DMSO-*d_6_*) δ 55.07, 69.44, 102.20, 113.28, 113.30, 115.98 (2C), 116.09 (2C, *J* = 22.7 Hz), 116.64, 119.81, 120.47, 123.33, 126.93 (2C), 127.01, 129.62, 129.72, 131.53, 132.17 (2C, *J* = 9.1 Hz), 133.28 (*J* = 3.1 Hz), 134.19, 134.81, 138.38, 139.08, 147.17, 155.26, 157.47, 159.36, 165.53 (*J* = 252.7 Hz), 187.10. Anal. calcd for C_32_H_25_FN_2_O_3_·1.4HCl: C 69.16, H 4.80, N 5.04; found: C 69.01, H 4.79, N 4.99.


**(*E*)-3-{4-{{4-[(4-Chlorobenzyl)oxy]phenyl}amino}quinolin-2-yl}-1-(4-fluorophenyl)prop-2-en-1-one *(*5g*)***


From **3** and 4-[(4-chlorobenzyl)oxy]aniline, as described for **4a**: Yield 87% an orange solid. Mp 179–180 °C. ^1^H NMR (400 MHz, DMSO-*d_6_*) δ 5.15 (s, 2H, CH_2_), 7.11–7.13 (m, 3H, Ar-H and 3-H), 7.34–7.43 (m, 4H, Ar-H), 7.47–7.57 (m, 5H, Ar-H and 6-H), 7.62 (d, 1H, *J* = 15.6 Hz, CH = CH), 7.72–7.76 (m, 1H, 7-H), 7.94 (d, 1H, *J* = 8.8 Hz, 5-H), 8.02 (d, 1H, *J* = 15.6 Hz, CH = CH), 8.13–8.17 (m, 2H, Ar-H), 8.40 (d, 1H, *J* = 8.4 Hz, 8-H), 9.00 (br s, 1H, NH). ^13^C NMR (100 MHz, DMSO-*d_6_*) δ 68.64, 101.51, 115.71 (2C), 115.96 (2C, *J* = 21.2 Hz), 119.16, 121.94, 125.20 (2C), 125.55, 128.46 (2C), 129.55 (2C), 129.95, 131.52 (2C, *J* = 9.1 Hz), 132.42, 133.02, 134.08, 136.18, 144.81, 148.75, 149.62, 152.92, 155.25, 165.06 (*J* = 250.1 Hz), 188.52. Anal. calcd for C_31_H_22_ClFN_2_O_2_·0.5HCl: C 70.61, H 4.31, N 5.31; found: C 70.67, H 4.39, N 5.15.


**(*E*)-3-{4-{{4-[(4-Fluorobenzyl)oxy]phenyl}amino}quinolin-2-yl}-1-(4-fluorophenyl)prop-2-en-1-one *(*5h*)***


From **3** and 4-[(4-fluorobenzyl)oxy]aniline, as described for **4a**: Yield 86% a yellow solid. Mp 254–255 °C. ^1^H NMR (400 MHz, DMSO-*d_6_*) δ 5.17 (s, 2H, CH_2_), 7.21–7.28 (m, 5H, Ar-H and 3-H), 7.42–7.48 (m, 4H, Ar-H), 7.54–7.57 (m, 2H, Ar-H), 7.72 (d, 1H, *J* = 15.6 Hz, CH = CH), 7.76–7.80 (m, 1H, 6-H), 8.02–8.07 (m, 1H, 7-H), 8.35–8.38 (m, 2H, Ar-H), 8.53 (d, 1H, *J* = 8.4 Hz, 5-H), 8.75 (d, 1H, *J* = 8.4 Hz, 8-H), 8.82–8.85 (d, 1H, *J* = 15.6 Hz, CH = CH), 10.98 (br s, 1H, NH), 14.70 (br s, 1H, HCl). ^13^C NMR (100 MHz, DMSO-*d_6_*) δ 68.86, 102.16, 115.33 (2C, *J* = 21.4 Hz), 115.98 (2C), 116.13 (2C, *J* = 21.7 Hz), 116.68, 120.54, 123.33, 126.96 (2C), 127.05, 129.80, 130.11 (2C, *J* = 8.4 Hz), 131.49, 132.19 (2C, *J* = 9.6 Hz), 133.04 (*J* = 3.0 Hz), 133.29 (*J* = 3.7 Hz), 134.23, 134.91, 139.12, 147.24, 155.26, 157.40, 161.85 (*J* = 242.4 Hz), 165.05 (*J* = 251.6 Hz), 187.13. Anal. calcd for C_31_H_22_F_2_N_2_O_2_·1.1HCl: C 69.89, H 4.38, N 5.26; found: C 69.93, H 4.23, N 5.23.

### 3.2. Biological Activity

#### 3.2.1. Antiproliferative Assay

Cancer cells (Huh-7, MDA-MB-231) and normal lung cells (MRC-5) were purchased from Bioresources Collection and Research Center, Taiwan. Cells were maintained in the same standard medium, grown as a monolayer in DMEM (Gibco, Miami, FL, USA), and supplemented with 10% fetal bovine serum (FBS) and antibiotics, i.e., 100 IU/mL penicillin, 0.1 mg/mL streptomycin and 0.25 μg/mL amphotericin. The culture was maintained at 37 °C with 5% CO_2_ in a humidified atmosphere. Cells (5 × 10^3^ cells/well) were treated with tested compounds for 72 h in the medium containing 10% FBS. Cell viability was quantitated with the use of sodium 3′-[1-(phenylamino-carbonyl)-3,4-tetrazolium}-bis(4-methoxy-6-nitro)benzene sulfonic acid hydrate (XTT) colorimetric assay (Biological Industries, Beit-Haemek, Israel). XTT labeling reagent (1 mg/mL) was mixed with electron-coupling reagent, following the manufacturer’s instructions, and 50 μL of the mixture was added directly to the cells. The plates were further incubated at 37 °C for 4 h. Color was measured spectrophotometrically in a microtiter plate reader at 492 nm and used as a relative measurement of viable cell numbers. After drug treatment, the viable cell number was compared to the solvent and untreated control cell number. This information was used to determine the percent of control growth as (Ab_treated_/Ab_control_) × 100, where Ab represents the mean absorbance (*n* = 3). The concentration that killed 50% of cells (IC_50_) was determined from the linear portion of the curve by calculating the concentration of the agent that reduced the absorbance in treated cells, compared to control cells, by 50%.

#### 3.2.2. Cellular ATP Assay

Cellular ATP was determined using the ATPlite luminescence reagent (PerkinElmer Life Sciences, Boston, MA, USA) [40,41]. According to the user instruction, the cell lysate of treated cells was reacted with ATPlite substrate for 5 min in the dark. Finally, the luminescence was detected by a luminometer (Berthold Technologies GmbH & Co., Bad Wildbad, Germany).

#### 3.2.3. Annexin V/7AAD-Apoptosis Assay

Annexin V/7AAD flow cytometry was a common method for detecting apoptosis [42]. The commercial product containing annexin V-FITC (1:1000) and 7AAD (1 μg/mL) [43] (Strong Biotech; Taipei, Taiwan) was applied to stain drug-treated cells for 1 h. Finally, these stained cells were conducted with flow cytometry (Guava easyCyte, Luminex, TX, USA).

#### 3.2.4. ROS Assay

The oxidative stress indicator, such as ROS, was selected to assess the impact of drug treatment on the change in oxidative stress. According to the manufacturer’s instructions, the ROS reacting probe 2′,7′-dichlorodihydrofluorescein diacetate (H_2_DCFDA) (Sigma-Aldrich) was applied to drug-treated cells [44,45]. Finally, these stained cells were conducted with flow cytometry.

#### 3.2.5. Caspase 3/7 Assay

Caspase 3/7 activation is the essential process for apoptosis, which was measured by the caspase-Glo^®^ 3/7 assay (Promega; Madison, WI, USA) [46,47,48], and its signal was read and recorded by a Luminometer (Berthold Technologies GmbH & Co., Bad Wildbad, Germany). The role of the induction of ROS affecting caspase 3/7 activity in drug-treated cells was evaluated by the 1 h pretreatment of 10 mM NAC [45,49,50].

### 3.3. Statistical Analysis

The significance was determined using JMP software 12 (SAS Institute Inc., Cary, NC, USA). Data containing non-overlapping notes exhibited significant differences.

## 4. Conclusions

In this study, we synthesized 4-anilinoquinolinylchalcone derivatives and evaluated them in vitro for their anticancer activities. These compounds were more potent in inhibiting Huh-7 and MDA-MB-231 cancer cell proliferation than the reference drug lapatinib. Among them, **4a** was active against the growth of MDA-MB-231 cells with an IC_50_ value of 0.11 μM without significant cytotoxicity to the normal MRC-5 cell line (IC_50_ > 20 μM). **4a** shows ATP depletion and apoptosis of breast cancer MDA-MB-231 cells, associated with ROS-dependent caspase 3/7 activation. Further studies on structural optimization are ongoing.

## Data Availability

Data are contained within the article.

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
