# Peer review of "Synthesis and Anticancer Evaluation of 4-Anilinoquinolinylchalcone Derivatives"

_ijms, 2023, doi:10.3390/ijms24076034_

Round 1

Reviewer 1 Report

The manuscript by Chih-Hua and Hsueb-Wei reported the synthesis, characterization of 4-anilinoquinolinylchalcone derivatives as new potential anticancer agents for the treatment of human cancers.

The originality of this work resides in the choice to synthesize a series of target 4-(benzyloxy)aniline-quinolinylchalcone hybrids. These molecules were then tested for their antiproliferative activities against the growth of Huh-7, MDA-MB-231 human cancer cell lines and normal lung cells (MRC-5).

Thus, compound 4a (E)-3-{4-{[4-(benzyloxy)phenyl]amino}quinolin-2-yl}-1-(4-methoxyphenyl) prop-2-en-1-one was found the highest cytotoxicity in breast cancer cells and low cytotoxicity in normal cells. Compound 4a causes ATP depletion and apoptosis of breast cancer MDA-MB-231 cells and triggers reactive oxygen species (ROS)-dependent caspase 3/7 activation. 4a is selectively active against the growth MDA-MB-231 with an IC50 value of 0.11 μM.

The products are well described and the supporting information is complete, the authors could have detailed the 13 C NMR part (Cquat, CH, CH2 and CH3)?

For all these reasons, this paper can be considered in Int. J. Mol. Sci. The manuscript reads well and contains few minor errors (see below).

-          Page 1, Chih-Hua Tseng ?

-          Page 3, Scheme 1 : why are the same operating conditions not used in the paper referred to in this article : Molecules, 2020, 25 (14), 3133 ? Acidic condition (6N) ? and time ?

-          Page 7, line190, TMS  = tetramethylsilane

Author Response

Dear Reviewer,

We would like to thank you for constructive comments and suggestions. Yours concerns and suggestions have been carefully taken into consideration. Accordingly, we have carefully revised the manuscript according to yours comments and addressed to all concerns by the reviewers. We have included point-to-point explanation to yours concerns. (Please see the PDF file.)

Reviewer 2 Report

The presented study of synthesizing the 4-Anilinoquinolinyl-chalcone analogs was a quite an interesting approach. Herein, authors utilised the scaffold of (E)-3-[3-(4-methoxyphenyl)quinolin- 2-yl]-1-phe-nylprop-2-en-1-one (as compound-1, as shown in fig 1) and lapatinib. 

Point 1. Although the authors are pointing out that the hybrid approach utilising chalcone struture of compound-1 and Phenylether amine substructure from Lapatinib, but didn't explain why only these substructures of compound-1 and lapatinib were considered and why not the other substructures from these molecules (from compound-1 and lapatinib). It seems that author considered those substructures that can be synthesized. The author needs to rationalize the reasoning behind of using such substructures from compound-1 and Lapatinib. For a suggestion, the author could utilize molecular modeling to find the significance of those substuctures that were considered in their study to the respective targets, (Such as, Lapatinib has EGFR kinase protein as a target).

Point 2. The choice of control: The author does use the Lapatinib as control drug for cytotoxicity evaluation of new compounds (as synthesized in the paper), but Lapatinib is mainly known to target kinase proteins however comparing its cytotoxicty with the new hybrid compounds need a proper justification as new compounds doesn't necessarily target EGFR. If that is the case, it would be nice to provide inhibitory kinase evaluation data for new compounds.

Point 3. As lapatinib was considered in cytotoxicity evalutaion why compound-1 was not considered as new compounds were resulted as the hybrid compounds from these molecules (molecules: lapatinib and compound-1).

Point 4. NMR spectroscopy:

4(a) The integration of 1H-NMR is confusing, especially the integration of individual protons, please address it accordingly.

4(b) Also, it would be nice to zoom-in those overlapped aromatic regions of 1H NMR, so that it would improve the readability of these spectra, as in current form these are highly congested.

4(c) 13C-NMR: Are there some missing in 13C signals. it would be nice if author check carefully.

Please check the 1H and 13C thoroughly.

Author Response

(The authors gave the same response as above.)

Reviewer 3 Report

The paper has addressed the synthesis and anticancer evaluation of 4-anilinoquinolinyl chalcone derivatives. The synthesized compounds have been evaluated for their biological activities via different techniques and reported to exhibit potent cytotoxic activity against MDA-MB-231 cell lines. The literature gap has been mentioned in the introduction as well as comparison of previous work with present work has also been explained. Here are few observations which require attention;

First paragraph of introduction (lines 41-47) does not contain any single reference!

The authors should explain SAR (Structure activity relationship) may be in the form of a paragraph.

Generally, the references are appropriate. However, authors name of few references has been ended with et al. such as ref 7. It is better to mention complete list of author names. 

§  Line 63 & 68 need to be rephrased.

§       Line 76… “were designed” should be replaced with “was designed”. Same applies for line 29.

       Line 142.. "exhibit structure" is not suitable. Use appropriate vocabulary according to sentence need.

§        Lines 170-172,175-176,188, 190, 403, 438 should be rewritten for better clarity and sentence composition.

§  The sign of IC50 value needs to be checked in line 449-450. 

  I would like to suggest authors to add molecular docking for the most potent compound to validate the results.

   English language of the manuscript should be improved.

Line 457; is it minister or ministry?

Author Response

(The authors gave the same response as above.)

Round 2

Reviewer 2 Report

The authors address most of the comments.

Specific to comment-1, 

Comment 1. However, the authors point out that the hybrid approach utilizing chalcone structure of compound-1 and Phenylether amine substructure from Lapatinib, but didn't explain why only these substructures of compound-1 and lapatinib were considered and why not the other substructures from these molecules (from compound-1 and lapatinib). It seems that author considered those substructures that can be synthesized. The author needs to rationalize the reasoning behind using such substructures from compound-1 and Lapatinib. For a suggestion, the author could utilize molecular modeling to find the significance of those substructures that were considered in their study to the respective targets (Such as Lapatinib has EGFR kinase protein as a target).

Authors responded, "The current selection of hybrid structure is based on to fragment-based drug design. The target structures are more drug-like with suitable molecular weight and the number of Hydrogen-bond donating/accepting groups."

Most of the fragment-based approaches utilize when the crystal structure of the target protein is unavailable. Therefore, designers often incorporate strategies involving other structural techniques, such as NMR-guided design, molecular modeling, or various chemical approaches.

For more information, 

Fragment-Based Drug Discovery: Advancing Fragments in the Absence of Crystal Structures. Cell Chemical Biology, 2019, 26 (1), 9-15.

Please elaborate with a proper justification of how the fragment-based drug design was applied, specifying the role of fragments taken from both the parent compounds (compound-1 and Lapatinib). This design study must be included in the current paper to improve the rationality of the paper. 

Author Response

Most of the fragment-based approaches utilize when the crystal structure of the target protein is unavailable. Therefore, designers often incorporate strategies involving other structural techniques, such as NMR-guided design, molecular modeling, or various chemical approaches.

For more information, 

Fragment-Based Drug Discovery: Advancing Fragments in the Absence of Crystal Structures. Cell Chemical Biology, 2019, 26 (1), 9-15.

Please elaborate with a proper justification of how the fragment-based drug design was applied, specifying the role of fragments taken from both the parent compounds (compound-1 and Lapatinib). This design study must be included in the current paper to improve the rationality of the paper. 

Ans: Thank you for the comment and suggestion. I am sorry that the rational design is not based on the fragment-based approaches, and therefore we have modified our manuscript and inserted the following sentences: “We considered the hybrid approach utilizing chalcone structure of compound-1 and Phenylether amine substructure from Lapatinib. The target structures are more drug-like with a shape similar to the compact cis-form of Combretastatin A4. They have also possessed suitable molecular weights and the number of hydrogen-bond donating / accepting groups according to Lipinski's rule of five.” in the revised manuscript Lines 75 – 84.

Reviewer 3 Report

The revised version is fine and can now be accepted for publication.

Author Response

Thank you for the comment.

Round 3

Reviewer 2 Report

The authors incorporate the necessary information.